# Hemorrhagic Cardiac Tamponade—An Unusual Threat in the COVID-19 Recovery Phase

**DOI:** 10.3390/medicina59010003

**Published:** 2022-12-20

**Authors:** Raluca Ungureanu, Cristian Cobilinschi, Ana-Maria Cotae, Raluca Darie, Radu Tincu, Sorin Constantinescu, Oana Avram, Horatiu Moldovan, Ioana Marina Grintescu, Liliana Mirea

**Affiliations:** 1Department of Anesthesiology and Intensive Care, Clinical Emergency Hospital Bucharest, 014461 Bucharest, Romania; 2Department of Anesthesiology and Intensive Care II, Carol Davila University of Medicine and Pharmacy, 050474 Bucharest, Romania; 3Department of Clinical Toxicology, Carol Davila University of Medicine and Pharmacy, 050474 Bucharest, Romania; 4Department of Radiology, Carol Davila University of Medicine and Pharmacy, 050474 Bucharest, Romania; 5Department of Radiology, Victor Atanasiu National Aviation and Space Medicine Institute, 010825 Bucharest, Romania; 6Department of Cardiovascular Surgery, Clinical Emergency Hospital Bucharest, 014461 Bucharest, Romania; 7Department of Cardiovascular Surgery, Carol Davila University of Medicine and Pharmacy, 050474 Bucharest, Romania; 8Academy of Romanian Scientists, 54, Spl. Independentei, 050711 Bucharest, Romania

**Keywords:** cardiac tamponade, coronavirus, COVID-19, pericarditis, pericardial involvement, critical care, cardiovascular surgery

## Abstract

Cardiac tamponade is a rare presentation in patients with COVID-19, which may be induced by the associated exacerbated inflammatory response. The onset of cardiac tamponade may be concomitant with the acute phase of the disease or may develop subsequently as a new health condition secondary to the disease. We report four cases of cardiac tamponade that occurred late after the acute phase of the disease. One of them may be considered a post-acute complication of the disease, and three of them may be classified as a new health condition induced by COVID-19. Only two cases had a history of severe respiratory distress due to COVID-19. In all four cases, pericardiocentesis was imposed, and surprisingly, in every case, hemorrhagic fluid was evacuated. In this case, series, immune-mediated etiology is supported by histopathological results, where the main identified feature was fibrous pericarditis with inflammatory infiltrate. Only one patient included in this report died, and three of them were discharged after anti-inflammatory treatment was initiated.

## 1. Introduction

Since March 2020, when World Health Organization declared the coronavirus disease 2019 (COVID-19) pandemic, an increased number of reports were dedicated to multisystemic manifestations of the disease [1]. The lungs are considered not only the main hotspot of the disease but also the center from which it is transmitted to the other organs [2]. Subsequently, a variety of early and late extrapulmonary complications induced by the exacerbated inflammatory response were reported [1,3].

A diverse spectrum of cardiac manifestations have already been described, including stress cardiomyopathy, myocarditis, myocardial ischemic events, and arrhythmias; however, very few papers have been dedicated to pericardial involvement [4,5]. Cardiovascular involvement may be induced through a direct effect of SARS-CoV-2, but also indirectly through oxidative stress and increased cytokines production [6]. Considering that endothelial cells play a central role in COVID-19 pathogenesis, the mechanisms of pericardial involvement, which is a relatively avascular structure, are still to be elucidated [4]. Molecular mimicry, an immune-mediated mechanism, was also considered to be part of COVID-19’s extrapulmonary involvement [7].

Although pericardial effusion is relatively prevalent, acute pericarditis and cardiac tamponade are rarely reported [8]. Cardiac tamponade emergence is reported to be variable according to the initial SARS-CoV-2 infection [9]. Taking into account that cardiac tamponade is a life-threatening condition that requires prompt intervention, increasing the awareness regarding this COVID-19 complication is mandatory. We further report four cases of cardiac tamponade.

## 2. Case Series

### 2.1. Case 1

A 69-year-old female patient with a previous medical history including severe heart failure (NYHA Class IV) and secondary pulmonary arterial hypertension was admitted to our unit with shortness of breath aggravated in the last 24 h. She was unvaccinated against COVID-19. Three weeks before admission, she was discharged from another hospital, where she was treated for a moderate form of COVID-19 pneumonia. Thromboprophylaxis with low-molecular-weight heparin (LMWH) was administered during hospitalization. Real-Time Polymerase Chain Reaction (RT-PCR) SARS-CoV-2 on readmission was negative. Focused assessed transthoracic echocardiography (FATE) performed in the emergency room revealed normal left ventricular ejection fraction (LVEF = 50%), circumferential pericardial effusion (maximal diameter around right cardiac chambers, 30 mm; apex, 16 mm; and posterior of the left ventricle, 21 mm), collapse of right cardiac chambers, paradoxical interventricular septal motion, and important bilateral pleural effusion (right pleura 75 mm/left pleura 36.5 mm). Chest CT scan also revealed circumferential pericardial effusion (Figure 1). Considering the rapid deterioration of her respiratory and hemodynamic state, she was directly transferred to the operating theater (OR), where subxiphoid pericardiocentesis and pleural drainage were performed under general anesthesia. An extended image of lab results is presented in Table A1. A total of 400 mL of hemorrhagic fluid was evacuated from the pericardium and 1000 mL of serous fluid from the pleura. Samples from both areas were sent for biochemical and microbiological analysis (see Table A2). In the postoperative period, she was transferred to the intensive care unit (ICU), intubated and under minimal vasopressor support (Noradrenaline = 0.08 μg/kg/min). Twelve hours, later a weaning trial was performed, and the patient was safely extubated. The minimal vasopressor requirements persisted in order to maintain a mean arterial pressure > 65 mmHg. On day 2, control transthoracic echocardiography (TTE) revealed an improved ventricular function with an LVEF = 55%. On day 5, TTE re-evaluation revealed increased pericardial thickness (≅1 cm), respirophasic ventricular septal shift, and inferior vena cava dilatation. Ibuprofen (600 mg TID) and colchicine (1 mg daily) treatment was initiated, and two days later, she was transferred to the ward.

### 2.2. Case 2

A 70-year-old female patient presented to our hospital for hypoxemic respiratory insufficiency due to severe SARS-CoV-2 pneumonia. She had a medical history of an atrio-ventricular block and rheumatoid arthritis. She was unvaccinated against COVID-19. She was admitted to the ICU, where high-flow-nasal-oxygen (HFNO) therapy was initiated (flow = 60 L/min and inspiratory oxygen fraction (FiO_2_) = 0.8). Thromboprophylaxis with LMWH was also started. An extended image of lab results is presented in Table A1. TTE from admission revealed a normal systolic function (LVEF = 55%) and circumferential pericardial effusion (lateral 13 mm and posterior from LV 10 mm). Two weeks later, her RT-PCR SARS-CoV-2 was negative. During this period, her respiratory function slowly improved, but weaning from the HFNO was not possible. On day 17, her general condition deteriorated, with shortness of breath despite an increase in HFNO parameters, circulatory failure (Mean arterial pressure (MAP) < 45 mmHg), and tacycardia (125 bpm). A rapid FATE evaluation confirmed cardiac tamponade diagnostics (diastolic right ventricular collapse and lack of inferior vena cava (IVC) collapsibility). Chest CT scan revealed increased pericardial effusion volume (posterior from left ventricle 25 mm), but also increased mediastinal adenopathy (Figure 2). She was directly transferred to the OR, where pericardiocentesis was performed and 750 mL of hemorrhagic fluid was drained. Samples from pericardial fluid were sent for biochemical and microbiological analysis (see Table A2). Postoperatively she was readmitted to the ICU, where weaning from mechanical ventilation succeeded on the next day and HFNO therapy was reinstated (Flow = 60 L/min, FiO_2_ = 0.8). After 48 h, another chest CT scan was performed, indicating a moderate right pleural effusion but no signs of pericardial effusion. After pleural drainage (1500 mL serous fluid), respiratory dysfunction slowly improved, and the patient was weaned from HFNO therapy. On day 7, treatment with ibuprofen (600 mg TID) and colchicine (1 mg daily) was initiated. Moreover, on day 28, she could be transferred to the ward with stable vital signs.

### 2.3. Case 3

A 66-year-old female patient was transferred to our unit from a COVID-19-designated hospital with clinical features of obstructive shock due to a cardiac tamponade. One week before, she had been diagnosed with severe pneumonia due to SARS-CoV-2 infection. No previous vaccination against COVID-19 was reported. On admission to the emergency room, the following were found: tachycardia—125 bpm; hypotension—87/45 mmHg; cold skin; hypoxemic respiratory insufficiency under invasive mechanical ventilation; and anuria. An extended image of lab results are presented in Table A1. Chest CT scan on admission indicated circumferential pericardial effusion (maximal thickness—48 mm) and extended pulmonary consolidation (Figure 3). Pericardiocentesis was performed under general anesthesia, and a total of 600 mL of hemorrhagic fluid were drained. Samples from the pericardial fluid were sent for biochemical and microbiological analysis (see Table A2). Intraoperatively, the patient was hemodynamically unstable, and vasopressor support was promptly initiated (mean dose of Noradrenaline = 0.18 μg/kg/min). Postoperatively, she was admitted to the ICU, where hemodynamic and respiratory dysfunctions were aggravated and additional support was raised. Unfortunately, on day 5, the patient died with multiple organ failure.

### 2.4. Case 4

A 40-year-old male patient presented to our hospital for hypoxemic respiratory dysfunction and oliguria. No significant medical history was reported, except for mild SARS-CoV-2 pneumonia three weeks before. No specific treatments were administered during this period. RT-PCR SARS-CoV-2 was negative, and he was not vaccinated against COVID-19. TTE on admission showed a decreased LV function (LVEF 20%) and a moderate pericardial effusion (maximal thickness—15 mm). A thoracic CT scan confirmed the circumferential pericardial effusion (Figure 4). An extended image of lab results is presented in Table A1. Forty-eight hours later, respiratory dysfunction worsened (oxygen saturation < 80%, dyspnoea, tachypnoea). He was transferred to the ICU, where HFNO therapy (Flow = 60 L/min, FiO_2_ = 85 %) was initiated, and a new TTE was performed, indicating cardiac tamponade development (increased pericardial fluid volume—35 mm; paradoxical interventricular septal motion; and IVC dilatation—24 mm). Meanwhile, signs of obstructive shock (tacycardia—135 bpm; MAP = 40 mmHg) developed, and emergency pericardiocentesis was performed under general anesthesia. Five hundred milliliters of hemorrhagic fluid was evacuated, and samples were sent for biochemical and microbiological examination (see Table A2). In the postoperative period, he was transferred to the ICU, where hemodynamic dysfunction persisted with a high demand for vasopressor (noradrenaline 1.5 μg/kg/min). TTE re-evaluation indicated an improved LV function (LVEF = 50%), persistence of paradoxical interventricular septal motion, and minimal pericardial fluid. Next, cardiovascular dysfunction improved slowly, and forty-eight hours later, a weaning trial was initiated and he was safely extubated. On day 4, treatment with Ibuprofen (600 mg TID) and colchicine (1 mg daily) was started, and afterwards the patient was transferred to the ward.

## 3. Discussion

Although there are some available data regarding COVID-19 pericardial involvement, the frequency and severity of this condition are not well described [9]. Nevertheless, acute pericarditis and cardiac tamponade are rarely reported [10,11]. Ghantous et al. reported that from 750 evaluated patients, 75 presented pericardial effusion and only 17 patients were diagnosed with acute pericarditis. None of them evolved into cardiac tamponade [8]. Moreover, most of the available reported cases are associated with the early course of the disease and concomitant positive SARS-CoV-2 RT-PCR detection [3,11,12].

We report four cases of cardiac tamponade developed with a variable onset time (mean period 15.7 days) after COVID-19 pneumonia onset. Only one case (Case 3) from our series still presented a positive SARS-CoV-2 RT-PCR detection at the time of cardiac tamponade onset. Farina et al. also reported negative nasopharyngeal swabs and blood viral RNA detection when cardiac tamponade manifested [13].

Although it has been reported that male patients have a higher risk of acute pericarditis and tamponade, in our case series, three out of four patients were females [14]. Hakmi et al. reported three male patients who developed cardiac tamponade with concomitant COVID-19 infection, as well as Sauer et al., who presented a case series with two out of three male patients with acute pericarditis and COVID-19 [3,12]. Considering the lack of epidemiological data regarding gender risk differences in patients with COVID-19-related cardiac tamponade, no remarks regarding a particular feature of this report can be made.

The definition of the post-acute COVID-19 period has no generally accepted definition; however, most authors estimate that this period may start three weeks after the first clinical symptoms of COVID-19 occur [15]. Clinical data also concur with the laboratory findings, considering that the mean duration of a positive polymerase chain reaction is approximately 24 days [16]. According to Cochrane Rehabilitation REH-COVER (Rehabilitation COVID-19 Evidence-based Response) action, after the acute phase, the course of the disease may either be extended with a post-acute phase or may evolve with a late onset of a new health condition secondary to COVID-19 [17]. In our case series, three patients (Cases 1, 2, and 4) started to develop typical signs of cardiac tamponade three weeks after respiratory symptoms evolved, corresponding to a late phase of the disease. For Case 3, the time until cardiovascular symptoms emerged is calculated from her first hospital admission, taking into account that the previous history is unavailable (the patient was living alone). As a result, Case 3 may be classified as a post-acute manifestation of COVID-19.

COVID-19 extrapulmonary lesions, including pleural effusion, were mostly associated with a severe course of the disease, worse pulmonary disease, and right ventricle dysfunction [8,10]. Brito et al. also reported that mild or moderate forms of COVID-19 may also be associated with pericardial involvement, including pericardial effusion. Their imaging study revealed that 19 out of 48 athletes with mild or moderate forms of the disease developed late in recovery-phase pericardial effusion [18]. Before cardiac tamponade onset, only two patients from our case series (Cases 2 and 3) had a history of a severe form of COVID-19, with both being admitted to the ICU. Cases 1 and 4 had a past history of a mild form of the disease, characterized by sore throat, mild fiber, and runny nose.

Cardiac involvement was also associated with influenza infection, often only with a mild course of the disease [19]. Different pericardial conditions with variable severity have also been reported in patients with influenza infection, including acute pericarditis and cardiac tamponade with obstructive shock [19].

The incriminated physiopathology mechanisms involved in COVID-19-related cardiac complication are variable according to the stage of the disease [7]. In our case series, already classified as post-acute or late-onset of a new disease complication, the possible implicated physiopathology mechanisms may include chronic inflammatory response against a persistent viral pool or molecular mimicry [7]. In this study, immune-mediated pathogenesis is also supported by histopathological results, where the main identified feature was fibrous pericarditis with inflammatory infiltrate.

Although in all four cases, elevated levels of high-sensitive cardiac troponin (hs-cTn) were detected, the peak values were only mildly to moderately increased. Considering that increased troponin levels are not exclusively associated with coronary heart disease, in this reported case series, mildly elevated hs-cTn may be interpreted as an additional marker of acute pericarditis [20].

Pericardiocentesis in all four reported cases revealed hemorrhagic pericardial fluid, indicating COVID-19 as an additional cause of hemorrhagic pericardial effusion. Providing that previously only Coxsackie virus infection was associated with hemorrhagic pericardial effusion, according to our case series, SARS-CoV-2 infection may also be a potential causal factor [21]. Similar findings were also reported in other published case reports [3,22].

Considering the current recommendation of the European Society of Cardiology, non-steroidal anti-inflammatory drugs (NSAIDs) and colchicine therapy were initiated after pericardiocentesis in three of the presented cases [14]. Weighing between the risk of rebleeding and cardiac tamponade recurrence, ibuprofen therapy initiation was decided under strict hemodynamic monitoring. In all three cases, evolution was favorable with no further complications.

### Limitations

Our study is limited by the limited number of cases included. None of the presented cases had RT-PCR testing for SARS-CoV-2 in the pericardial fluid or tissue; however, available data derived mostly from autopsy series revealed that the viral genome could not be identified in the pericardial tissue in none of the analyzed cases [23].

## 4. Conclusions

Cardiac tamponade is a rare complication of COVID-19, which may occur even during the recovery phase, long after the respiratory dysfunction is alleviated. This case series may demonstrate that cardiac tamponade may occur as a post-acute complication, as well as a late-onset a new health condition due to COVID-19. As a result, screening echocardiography should be considered for patients with a recent history of COVID-19 infection and sudden cardiovascular dysfunction. Nevertheless, proper biochemical monitoring including hs-cTn and CRP should also be performed for every patient with pericardial effusion in the post-acute phase of COVID-19 infection.

Immune-mediated etiology of COVID-19-related pericardial involvement may be supported by histopathological analysis, where the main identified feature was fibrous pericarditis with inflammatory infiltrate. Another feature of the cardiac tamponade induced by COVID-19, which was reported in this study, was the hemorrhagic feature of the evacuated pericardial fluid.

## Figures and Tables

**Figure 1 medicina-59-00003-f001:**
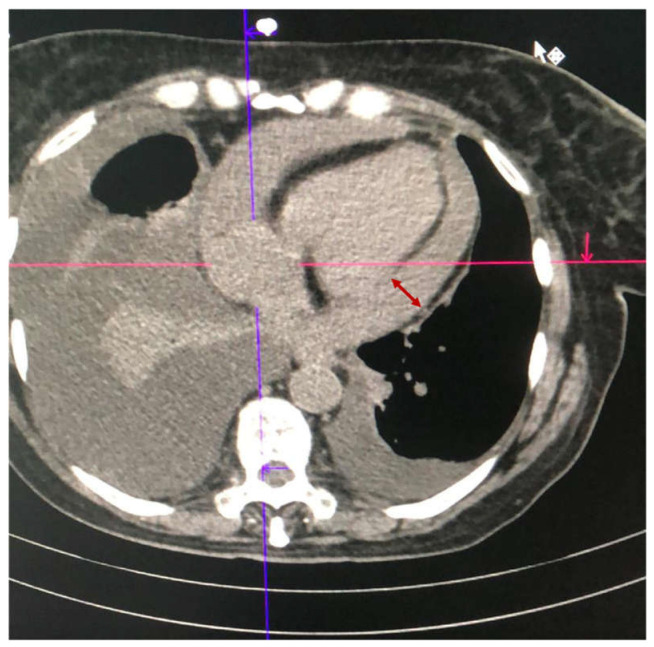
Thoracic CT-scan showing pericardial effusion (*Case 1*).

**Figure 2 medicina-59-00003-f002:**
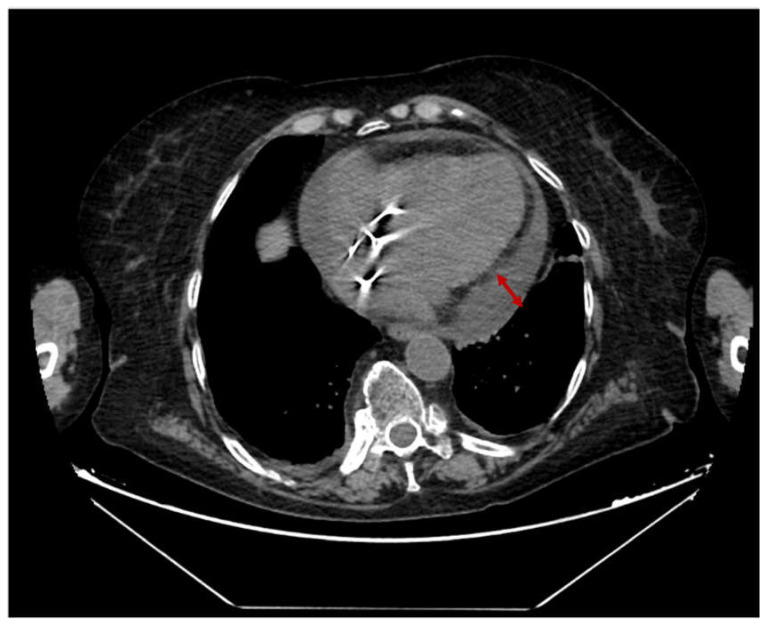
Thoracic CT-scan showing pericardial effusion (*Case 2*).

**Figure 3 medicina-59-00003-f003:**
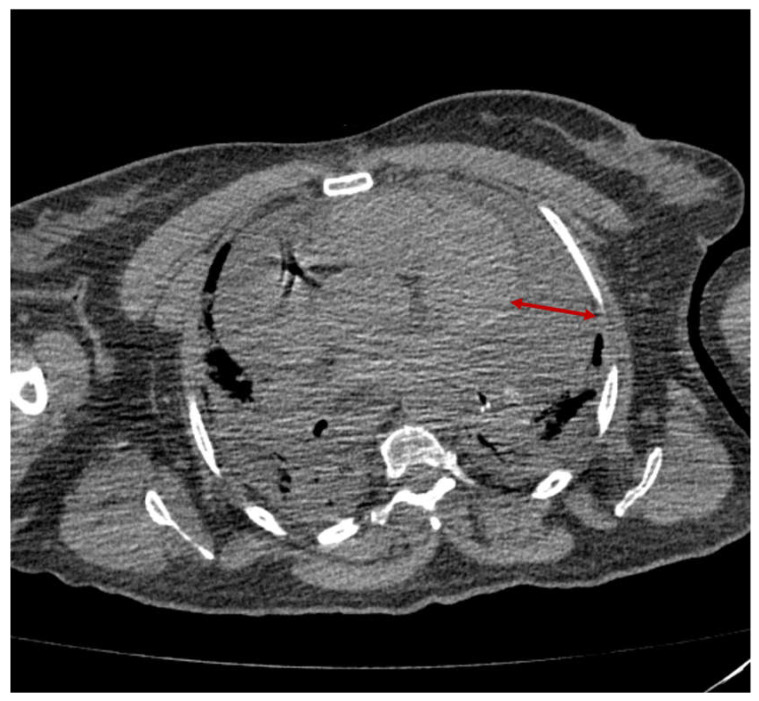
Thoracic CT-scan showing circumferential pericardial fluid (*Case 3*).

**Figure 4 medicina-59-00003-f004:**
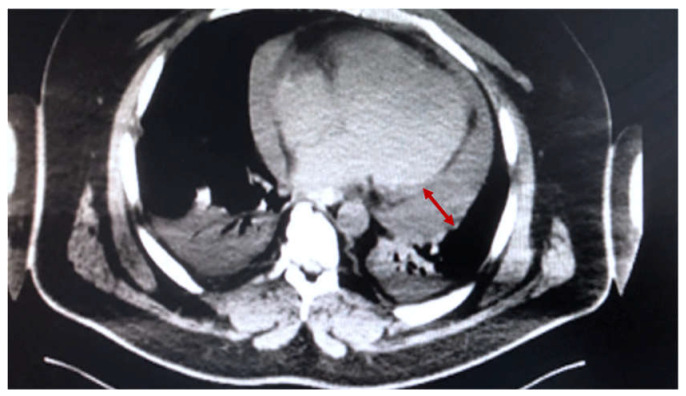
Thoracic CT scan showing circumferential pericardial fluid (*Case 4*).

## Data Availability

Not applicable.

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
