# Peer review of "Hemorrhagic Cardiac Tamponade—An Unusual Threat in the COVID-19 Recovery Phase"

_medicina, 2022, doi:10.3390/medicina59010003_

Round 1
Reviewer 1 Report
In this case series the authors from Romania described 4 patients ( 3 female) who developed pericarditis and/or myopericarditis subacutely , around 2 weeks after COVID 19 acute infection. They cases are interesting and worth publishing, however some improvement in mandatory in my opinion to enhance the quality for the readers.
Abstract
1. Line 28- remove case 2&3
2. Line 32- remove case 3
Introduction
3. Line 45- reference 5 is about ANCA vasculitis in COVId 19. I think better references would be the following : https://www.mdpi.com/2077-0383/11/8/2219 and https://www.mdpi.com/1999-4915/13/10/1997
Cases
4. Case 1- try to shorten deleting non essential information and adding pertinent one.
5. Was patient vaccinated against covid?
6. Was she treated with anticoagulation 3 weeks before and/or during post covid recovery period?
7. Ibuprofen is an interesting choice for treatment given the hemorrhage and risk of bleeding with all NSAIDs. This needs to be elaborated on in Discussion section
8. Case 2- same comments as for case 1: vaccination status, anticoagulation, choice of therapy with ibuprofen and case needs to be trimmed
9. Case 3- same questions as for previous cases
10. Case 4- same as for above cases
Discussion
11. Table 2 and 3 could be combined in one table
12. I would add levels of troponin, CRP- these are used to diagnose pericarditis and myopericarditis in Table 1
13. Table 4 in my opinion is also not so important and informative for practicing clinician
14. Discussion must be more informative and educational. For example, 3/4 patients are females and possible gender predilection for cardiac complication for covid should be briefly discussed.
15. Furthermore- please compare and contrast pericardial involvement in covid and compare with other common virus- influenza pericarditis ( see https://www.mdpi.com/2077-0383/11/14/4123)
Conclusion
16 . Suggestion for clinician should also be made in this section. For example, certain patients might benefit from focused ECHO screening? ongoing inflammatory markers monitoring CRP? troponin?
Author Response
Dear Reviewer,
Thank you very much for all your comments. I think that every recommendation made was very useful in order to improve my manuscript.
Point 1: 1. Line 28- remove case 2&3
- Line 32- remove case 3
Response 1: Done
Point 2: 3. Line 45- reference 5 is about ANCA vasculitis in COVId 19. I think better references would be the following : https://www.mdpi.com/2077-0383/11/8/2219 and https://www.mdpi.com/1999-4915/13/10/1997
Response 2: The reference was changes as recommended.
Point 3: Cases
- Case 1- try to shorten deleting non essential information and adding pertinent one.
- Was patient vaccinated against covid?
- Was she treated with anticoagulation 3 weeks before and/or during post covid recovery period?
- Ibuprofen is an interesting choice for treatment given the hemorrhage and risk of bleeding with all NSAIDs. This needs to be elaborated on in Discussion section
- Case 2- same comments as for case 1: vaccination status, anticoagulation, choice of therapy with ibuprofen and case needs to be trimmed
- Case 3- same questions as for previous cases
- Case 4- same as for above cases
Response 3: All non-essential information regarding medical history was deleted for all four cases. Vaccination status and thromboprophylaxis was added for all four cases. A new paragraph regarding NSAIDs administration was added in the discussion.
Point 4: C Discussion
- Table 2 and 3 could be combined in one table
- I would add levels of troponin, CRP- these are used to diagnose pericarditis and myopericarditis in Table 1
- Table 4 in my opinion is also not so important and informative for practicing clinician
- Discussion must be more informative and educational. For example, 3/4 patients are females and possible gender predilection for cardiac complication for covid should be briefly discussed.
- Furthermore- please compare and contrast pericardial involvement in covid and compare with other common virus- influenza pericarditis ( see https://www.mdpi.com/2077-0383/11/14/4123)
Response 4: Table 2 and 3 were combined and Table 4 was deleted.
Peak levels of high-sensitive cardiac troponin was included in Table 1.
Discussion section was enriched with paragraphs dedicated to gender related risk and similar features between pericardial conditions induced by COVID-19, respectively Influenza.
Point 5: Conclusion
16 . Suggestion for clinician should also be made in this section. For example, certain patients might benefit from focused ECHO screening? ongoing inflammatory markers monitoring CRP? troponin?
Response 5: A new paragraph where the role of FATE screening is underlined, was added.

Reviewer 2 Report
Medical community has learnt that COVID-19 is a multi-organ disease.
The authors of the manuscript has presented four cases of COVID-related pericarditis accompanied with cardiac tamponade.
The paper is well written, contains all necessary parts. It is appropriately illustrated with CT images.
There are some questions and comments to the authors:
- The cardiac tamponade is not equivalent to the pericardial effusion or pericarditis. It looks like the authors mostly discuss pericarditis related to COVID-19 n the manuscript.
- The cardiac tamponade is a medical emergency and signs of acute hemodynamics compromise are the main features of this syndrome. Most cases of pericardial effusions are not accompanied with the cardiac tamponade. In cases of 2 and 4 the presence of cardiac tamponade is not obvious from the description of patient’s status.
- Legends to all cases indicate the presence of pericardial effusion. What about signs of right heart chambers collapse and pulmonary congestion ?
- Tables 1-3 add very little to understanding of cardiac tamponade manifestations in these patients. Is it possible to add a table with hemodynamics parameters (HR, BP and main echocardiographic parameters) in patients included in this manuscript?
Author Response
Dear Reviewer,
Thank you very much for your positive comments and your recommendation.
As suggested I tried to solve all the suggested problems, as follows:
For every described case I emphasised the moment of cardiac tamponade development and more clinical data and TTE parameters were added (RV collapsibility, IVC diameter or septal movement).
Considering that the other reviewer recommended to delete some of the tables I included the above mentioned data inside the case description.

Round 2
Reviewer 1 Report
The authors have improved the paper significantly and I do not have any further comments. Congratulations!